# Cerebral blood flow during simulated central hypovolaemia in people with hypertension: does vertebral artery hypoplasia matter?

Sandra Neumann[1] , Jonathan C. L. Rodrigues[2], Lydia L. Simpson[1], Chris B. Lawton[3], Daniel Burden[3], Matthew D. Kobetic[1,3], Zoe H. Adams[1,4] , Katrina Hope[1], Julian F. R. Paton[5] , Hazel Blythe[1], Nathan Manghat[3] , Jill N. Barnes[6] , Angus K. Nightingale[1] , Mark C. H. Hamilton[3] and Emma C. Hart[1]

[1] *School of Physiology, Pharmacology and Neuroscience, University of Bristol, Bristol, UK*
[2] *Department of Radiology, Royal United Hospitals Bath NHS Foundation Trust, UK*
[3] *Department of Cardiology, University Hospitals Bristol and Weston NHS Foundation Trust, UK*
[4] *Cardiff School of Sport and Health Sciences, Cardiff Metropolitan University, Cardiff, UK*
[5] *Manaaki Manawa, The Centre for Heart Research, University of Auckland, Auckland, New Zealand*
[6] *Department of Kinesiology, Bruno Balke Biodynamics Laboratory, University of Wisconsin-Madison, Madison, WI, USA*

Handling Editors: Harold Schultz & Philip Ainslie

The peer review history is available in the Supporting Information section of this article (https://doi.org/10.1113/JP287786#support-information-section)

*The Journal of Physiology*

**Abstract figure legend** Participants with hypertension underwent magnetic resonance angiography (MRA) during central hypovolaemia induced by lower body negative pressure (LBNP). Participants were assigned to a group with vertebral artery hypoplasia (VAH; $n = 13$) or without VAH group ($n = 11$) post-acquisition. Phase-contrast MRA measured flow in the basilar artery (BA), internal carotid arteries (ICA), and the ascending aorta to measure cardiac output (CO). The ICA flow decreased during LBNP and was not different between groups. Total CBF and BA flow was decreased during LBNP in hypertensives without VAH but surprisingly was unchanged in patients with VAH. Blood pressure (BP) was reduced in the group without VAH only, whereas the rise in total peripheral resistance (TPR) was greater in the group with VAH. In summary, hypertensive patients without VAH may tolerate decreases in CBF, whereas patients with VAH evoke a greater systemic TPR response to preserve CBF.

The Journal of Physiology

**Abstract** Adults with hypertension have higher prevalence of vertebral artery hypoplasia (VAH), which is associated with lower resting cerebral blood flow (CBF). We examined whether VAH impacts the ability to regulate CBF during haemodynamic stress when cardiac output and blood pressure are lowered via body negative pressure (LBNP). Participants underwent magnetic resonance angiography (MRA) at 1.5T during LBNP at 0, −20 and −40 mmHg, and were assigned to VAH ($n = 13$) or without-VAH ($n = 11$) groups post-acquisition. Phase-contrast MRA measured flow in the basilar artery (BA), internal carotid arteries (ICA), and the ascending aorta to measure cardiac output (CO). The CO decreased during all levels of LBNP in both groups (LBNP main effect $P < 0.0001$), whereas MAP was reduced in the group without VAH only ($P = 0.0003$). BA flow was reduced during LBNP in the group without VAH ($P = 0.0267$ at −20 mmHg and $P < 0.0001$ at −40 mmHg) but was surprisingly unchanged in the group with VAH ($P > 0.05$ all levels LBNP). ICA flow decreased during LBNP ($P < 0.0001$) and was not different between groups. Total CBF decreased during LBNP in hypertensives without VAH ($P = 0.0192$ at −20 mmHg and $P < 0.0001$ at −40 mmHg) but was unchanged in patients with VAH ($P > 0.05$ at all levels of LBNP). Total peripheral resistance (TPR) increased during LBNP in both groups, but the rise was greater in the group with VAH (−20 mmHg; $P = 0.0129$, –40 mmHg; $P = 0.0016$). In summary, hypertensive patients without VAH may tolerate decreases in CBF, whereas patients with VAH evoke a greater systemic TPR response to preserve CBF.

(Received 7 October 2024; accepted after revision 28 January 2025; first published online 17 February 2025)

**Corresponding author** Emma C. Hart: School of Physiology, Pharmacology and Neuroscience, Biomedical Sciences Building, University of Bristol, BS8 1TD, UK. Email: Emma.hart@bristol.ac.uk

**Key points**

- Vertebral artery hypoplasia (VAH) is more common in hypertensive adults and is associated with lower resting cerebral blood flow (CBF), suggesting that VAH might impair the brain's ability to maintain cerebral blood flow during haemodynamic stress using lower body negative pressure.
- This study shows that hypertensive patients with VAH maintain CBF during body negative pressure, unlike those without VAH, who experience reductions in CBF. Patients with VAH show a greater rise in total peripheral resistance (TPR), suggesting a compensatory mechanism to maintain cerebral perfusion.
- The findings highlight that patients with VAH have an altered physiological response to hypovolaemia, where they may rely on systemic pressor responses to maintain perfusion of posterior brain territories in already hypoperfused circulation.
- This is important for understanding how VAH impacts cerebrovascular function in hypertensive patients and may influence clinical approaches to managing CBF in disease conditions.

## Introduction

Hypertension is a risk factor for the development of cerebrovascular disease and cognitive decline (Gottesman et al., 2014; Iadecola & Davisson, 2008). Patients with hypertension have a lower resting cerebral blood flow (CBF) per 100 g of tissue compared to normotensive adults (Warnert, Rodrigues et al., 2016). Additionally,

**Sandra Neumann** is a neuroscientist and clinical trial manager with experience working with patients, as well as a broad range of *in vivo* and *in vitro* techniques. Dr Neumann obtained her PhD at the University of Bristol which focused on interactions between pain and hypertension. Sandra's current research interests are how blood pressure regulation changes the sensation of pain.

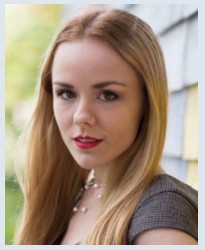

patients with hypertension have an impaired regional CBF response to metabolic challenge and cognitive tasks (Jennings et al., 2005). We have shown that the pre-valence of vertebral artery hypoplasia (VAH), a congenital anatomical variant leading to insufficient development of one or both of the vertebral vessels, is higher in patients with hypertension *versus* normotensive adults (Warnert, Rodrigues et al., 2016). This relationship also persists in young adults with hypertension (Manghat et al., 2022) and in hypertensive adults with previously repaired coarctation of the aorta (Rodrigues et al., 2019). The vertebral arteries are important in supplying the posterior brain territories, and vertebral hypoplasia is linked to poorer blood flow in these regions (Thierfelder et al., 2014). Importantly, our group also found that VAH was associated with resting cerebral hypoperfusion in patients with hypertension *versus* hypertensive patients with normal vertebral anatomy (Warnert, Rodrigues et al., 2016). Additionally, we also showed that people with VAH appear to have a larger systemic pressor response to a metabolic challenge in the occipital cortex (i.e. in response to neuronal activation via visual stimulus) compared to those without the variant (Warnert, Rodrigues et al., 2016). This suggests that hypertensive patients with VAH may rely on systemic cardiovascular responses to maintain or increase their CBF to meet metabolic demand.

During simulated hypovolaemia, middle cerebral artery (MCA) velocity, (commonly used as an index of cerebral blood flow) decreases when cardiac output is reduced (Brown et al., 2003; Neumann et al., 2019; Ogoh et al., 2005), and this can occur even before mean arterial pressure is lowered (Levine et al., 1994). A recent study directly quantifying CBF using magnetic resonance imaging (MRI) supports these studies, showing that cerebral blood volume in the large cerebral arteries is reduced during moderate levels of lower body negative pressure (LBNP, at −40 mmHg) (Whittaker et al., 2017). These findings suggests that under conditions where cardiac output is reduced, there may be a reduced ability to maintain CBF (Neumann et al., 2019). This is important because cardiac output decreases under several conditions, for example during general anaesthesia, orthostasis and dehydration. Interestingly, Bronzwaer et al. (2017) demonstrated that acute reductions in cardiac output were moderately linked to decreased CBF (estimated by MCA blood velocity) in young adults, but the relationship was steeper in older individuals, suggesting that with advancing age, there is an increased reliance on cardiac output to maintain cerebral perfusion.

Given that the cerebral circulation may be more sensitive to decreases in cardiac output in middle-aged individuals, combined with our recent work showing that hypertensive adults with VAH have lower CBF at rest (Warnert, Rodrigues et al., 2016), it is unclear if adults with hypertension and VAH are particularly sensitive to reductions in cardiac output. Additionally, our previous work indicated that hypertensive adults with VAH rely on systemic pressor responses to meet increased cerebral metabolic demand (Warnert, Rodrigues et al., 2016); thus, it is possible these patients (hypertension + VAH) may be at elevated risk of hypoperfusion in their posterior circulation during low to moderate levels of hypovolaemia. This could exacerbate an already 'hypoperfused' posterior circulation and could help to explain why posterior ischaemic strokes are more prevalent in people with VAH (Perren et al., 2007).

Therefore, the primary aim of this study was to establish whether blood flow in the basilar artery (BA) was impacted differently in hypertensive patients with VAH *versus* those without this anatomical variant, during central hypovolaemia induced by low to moderate levels of LBNP (−20 and −40 mmHg). We used the BA because (1) it is supplied by the left and right vertebral arteries, so reductions in flow in one vertebral artery may impact basilar flow and (2) the BA is important for brain-stem perfusion which regulates autonomic nerve activity and blood pressure. We hypothesised that hypertensive patients with VAH would have a larger reduction in BA blood flow during LBNP *versus* patients without the variant. Our additional aims were to: (1) assess whether the internal carotid artery (ICA) blood flow response to LBNP was different between patient groups; (2) examine the total CBF response during LBNP in these patient groups; and (3) determine whether changes in total peripheral resistance (TPR) during LBNP were different between groups.

## Methods

### Ethical approval

Following approval by the NHS Research Ethics Committee (15/SW/0176) and local R&D approval (Ref number: 2649), 26 participants with hypertension volunteered to take part in this study. Participants provided their written informed consent. All methods conformed to the *Declaration of Helsinki* (except for registration in a database as this was not a clinical trial), as well as local and UK national guidelines.

### Study population

Blood pressure status was confirmed by clinic and 24-h ambulatory monitoring in accordance with the European Guidelines for the management of arterial hypertension (Williams et al., 2018). Participants attended a screening visit, where they were asked to complete a health questionnaire (screening for previous diagnosis of any disease/condition), as well as a 12-lead ECG and urine

dipstick test (to rule out undiagnosed possible conduction abnormalities, including diabetes and kidney disease). Inclusion criteria were clinical diagnosis of hypertension. Patients had: (1) treated-controlled hypertension (taking one or more anti-hypertensive medications, a clinic blood pressure <140/90 mmHg and a daytime ambulatory blood pressure monitoring (ABPM) <135/85 mmHg), (2) treated-uncontrolled hypertension (taking one or more medications but clinic and daytime ABPM ≥140/90 or ≥135/85 mmHg, respectively or (3) untreated hypertension (taking no medications but has an elevated blood pressure; clinic and daytime ABPM ≥140/90 or ≥135/85 mmHg, respectively).

Participants were excluded if they were pregnant or had been diagnosed with any cardiovascular (excluding hypertension, but including non-benign conduction abnormalities, previous acute myocardial infarction, previous cardiac arrest, or other significant cardiopathy), respiratory (including chronic obstructive pulmonary disease, severe asthma), renal, cognitive or immunological and autoimmune disorders, diabetes, cancer and/or had a body mass index (BMI) >35 kg/m$^2$.

Participants were assigned to the group with VAH or the group with normal vertebral anatomy (i.e. without VAH) following diagnosis by a Consultant Radiologist (Warnert, Rodrigues et al., 2016). Our previous study showed a 57% prevalence of VAH in patients with hypertension; therefore, we expected a ~50%–60% rate of hypoplasia in the hypertensive population recruited for this study.

## Experimental protocol

**MRI protocol.** All participants underwent scanning using a 1.5T MRI (Siemans Avanto, Siemens Healthineers, Erlangen, Germany). A time-of-flight angiogram covering the superior parts of the internal carotid arteries, the vertebral artery confluence, the BA and the circle of Willis was acquired with a field of view of 200 mm, voxel size of 1.0 × 1.0 × 5 mm with a TR of 23 ms and TE of 7 ms, to characterise cerebral anatomy. Next, cardiac-gated phase contrast MR angiography was completed to measure blood flow in the basilar and internal carotid arteries. This was measured in the transverse plane, perpendicular to the internal carotid arteries at the level of the BA. Blood flow in the vertebral arteries was not measured due to methodological limitations with measuring flow in small hypoplastic vessels. The acquisition of the phase contrast angiography had a field of view of 250 mm, slice thickness of 5 mm, matrix size of 256 × 100 pixels, voxel size of 1.0 × 1.0 × 5 mm, a TR of 10.95 ms and a TE of 2.61 ms. We acquired 100 phase reconstructions at a maximum velocity encoding, VENC = 150 cm/s.

Aortic flow was acquired by phase contrast angiography in the ascending and descending aorta at the level of the main pulmonary trunk, to measure cardiac output at rest and during LBNP. The flows were performed during free breathing.

**Lower body negative pressure protocol.** Following 10–15 min of supine rest in the scanner (for acclimatisation), central hypovolaemia was induced by graded LBNP. Participants lay in a chamber sealed at the iliac crest (Hamilton et al., 2021). Participants were instructed to avoid moving during the protocol. The protocol consisted of graded LBNP at 0, −20 and −40 mmHg (Hamilton et al., 2021). Once the desired level of pressure was reached inside the chamber (measured via a barometer) the chamber pressure was maintained at each level of LBNP for 1 min before imaging. Phase contrast acquisition took ~4 min and the cardiac scan took ~3 min; therefore, participants remained at each level for LBNP for ~8 min. At each transition between levels of LBNP there is movement of the position of the head and the thorax, therefore, at the start of each new level of LBNP, images were re-localised to obtain blood flow measurements from the same anatomical position in the blood vessels of interest.

## Experimental measurements

**Haemodynamic monitoring.** During each scan, blood pressure and heart rate measurements were taken from the brachial artery using an automated sphygmomanometer (Omron). Measurements were taken at 1, 3 and 5 min during the phase contrast acquisitions. Expired $CO_2$ was measured (Capstar100, CWE Inc., PA, USA) and sampled at 200 Hz (PowerLab, AD Instruments) to obtain measures of partial pressure of end-tidal carbon dioxide ($P_{ET,CO_2}$).

## Data analysis

**MR image analysis.** The arteries were contoured in each image of reconstructed flow. The inter-observer variability between two independent observers was assessed by inter-class correlation in SPSS v23. Mean flow velocity and total flow in the left and right ICA and BA were quantified using semi-automated Siemens software (Argus, Siemens Healthineers, Germany). Total CBF was calculated as the sum of blood flow in left and right ICA and BA. Mean arterial blood pressure (MAP) was calculated as DBP + 1/3 × (SBP − DBP). The TPR was calculated as MAP (mmHg)/cardiac output (l/min). Cardiac output was measured via blood flow in the ascending aorta and cardiac index was calculated as cardiac output/body surface area. The vascular conductance in each vessel was calculated as blood flow (ml/min)/MAP (mmHg) (Zhang et al., 2004). Total cerebral vascular conductance (CVC) was calculated as the total CBF/MAP.

An experienced radiologist completed analysis of cerebral angiograms for the diagnosis of VAH. Source data were reviewed in three orthogonal multiplanar reformatted planes with cross-referencing of images. Maximum intensity projection images were generated and reviewed. All MR images were reviewed on a dedicated workstation (Insignia Medical Systems, UK). The visualised V2, V3 and V4 segments were analysed. VAH was defined as a diameter <2 mm uniformly throughout the vessel, and not if only a focal narrowing was presented suggestive of atherosclerotic steno-occlusive disease, as previously described (Park et al., 2007; Warnert, Rodrigues et al., 2016).

### Statistical analysis

All analyses were blinded so that the data analyser did not know the level of LBNP at which the data were acquired. Statistical analyses were conducted using GraphPad Prism (V9). Normality of the data was assessed by D'Agostino-Pearson test of normality. An unpaired Students $t$ test was completed to assess differences in demographics between groups. A two-way mixed factor analysis of variance (ANOVA) was completed to compare the effect of LBNP on systemic and cerebral haemodynamic variables between hypertensive patients without VAH and patients with VAH. Multiple comparisons tests were completed to assess the effect of the level of LBNP and the effect of group responses to each level of LBNP (Bonferroni multiple comparisons) on systemic and cerebral haemodynamic variables. Pearson's correlation coefficients were used to assess the relationship of BMI or changes in $P_{ET,CO_2}$ to the CBF response during LBNP. Finally differences in the number of medications prescribed between groups were analysed using a Fisher's exact test. The data are reported as mean and standard deviation or median and interquartile range where relevant. Alpha was set at 0.05. Where $P$ values are between 0.05 and 0.06 we have also provided the effect size (Cohen's $d$).

### Results

Two patients were not included in the analyses the angiogram images acquired were poor and thus the presence of VAH could not be determined. Thirteen patients were assigned to the group with VAH, and 11 were assigned to the group with normal vertebral anatomy (showing a 54% prevalence of VAH in this hypertensive population).

Blood flow measurements in the left ICA and ascending aorta were successfully obtained in 12 out of the 13 participants in the group with VAH. Therefore, data used to calculate left ICA, total CBF, cardiac output and TPR in the group with VAH are from 12 participants.

### Demographics

There were no differences in age, height, clinic and ambulatory blood pressures, cardiac output, cardiac index and TPR at rest between groups (Table 1). There was no difference in the number of anti-hypertensive medications prescribed between groups (Fishers exact, $P = 0.387$). In the group without VAH; two participants were untreated for hypertension and two were treated but had uncontrolled BP. In the group with VAH; five participants were untreated for hypertension and three had treated but uncontrolled BP. There was a trend for body mass and BMI ($P = 0.051$) to be lower in the group with VAH, where Cohen's $d$ was 0.83 for body mass and 0.87 for BMI, indicating a large effect size. There was no correlation of BMI to any cerebral measurements at rest or during each level of LBNP (Appendix Table A1).

### Resting cerebral blood flow

Blood flow in the BA at rest (i.e. at 0 mmHg of LBNP) was lower in the patients with VAH ($87 \pm 47$ ml/min) compared to the group without VAH ($153 \pm 33$ ml/min, Fig. 1). As resting MAP was similar between groups, BA vascular conductance at rest was lower in the patients with VAH ($0.81 \pm 0.47$ ml/min/mmHg) compared with those with normal anatomy ($1.44 \pm 0.33$ ml/min/mmHg). Blood flow in the left and right ICA was similar between groups (Fig. 2; VAH; $167 \pm 61$ and $235 \pm 65$ ml/min, *vs.* without VAH; $216 \pm 46$ and $205 \pm 60$ ml/min). Additionally, conductance in the left and right ICA was similar between groups (left ICA, VAH; $2.04 \pm 0.47$ *vs.* no VAH; $1.57 \pm 0.57$ ml/min/mmHg, $P = 0.074$ and right ICA, VAH; $1.94 \pm 0.60$ *vs.* no VAH; $2.20 \pm 0.68$ ml/min/mmHg).

Consistent with our previous reports, total CBF at rest was lower in the patients with VAH ($478 \pm 131$ ml/min) compared to patients without VAH ($573 \pm 114$ ml/min; Fig. 3). There was no group effect on total cerebral vascular conductance (no VAH; $5.37 \pm 1.30$ *vs.* VAH; $4.36 \pm 1.21$ ml/in/mmHg).

### Systemic haemodynamic responses to LBNP

Fig. 4 shows the systemic responses to LBNP and the ANOVA $P$ values (Appendix Table A2 for mean $\pm$ SD data and all statistical comparison data). HR increased during LBNP, but this occurred only at $-40$ mmHg and this response was not different between groups. MAP decreased with LBNP, but this appeared only in the group

**Table 1. Characteristics of hypertensive (HTN) patients with and without vertebral artery hypoplasia (VAH)**

| | HTN without VAH (*n* = 11) | HTN with VAH (*n* = 13) | *P* value |
|---|---|---|---|
| Male/female (*n*) | 5/6 | 7/6 | — |
| Age (years) | 52 ± 11 | 58 ± 7 | 0.097 |
| Height (cm) | 174 ± 11 | 171 ± 13 | 0.540 |
| Weight (kg) | 86.3 ± 10.2 | 77.6 ± 10.8 | 0.058 |
| BMI (kg/m$^2$) | 28.4 ± 2.3 | 26.5 ± 2.1 | 0.051 |
| Clinic BP & HR | 138 ± 1287 ± 8102 ± 873 ± 10 | 151 ± 1488 ± 8108 ± 1174 ± 10 | 0.0270.7040.1490.743 |
| SBP (mmHg) | | | |
| DBP (mmHg) | | | |
| MAP (mmHg) | | | |
| HR (beats/min) | | | |
| ABPM (daytime) | 133 ± 783 ± 8102 ± 771 ± 11 | 139 ± 1186 ± 9107 ± 1170 ± 7 | 0.1130.3780.1420.804 |
| SBP (mmHg) | | | |
| DBP (mmHg) | | | |
| MAP (mmHg) | | | |
| HR (beats/min) | | | |
| Cardiac MRI | 5.3 ± 1.32.60 ± 0.6168 ± 1421.7 ± 7.4 | 4.5 ± 0.782.33 ± 0.2966 ± 1225.2 ± 6.4 | 0.0970.2050.7210.235 |
| Cardiac output (l/min) | | | |
| Cardiac index (L/min/m$^2$) | | | |
| Heart rate (beats/min) | | | |
| TPR (mmHg/l/min) | | | |
| Anti-hypertensive | 824694599279 | 62318230080 | 0.386* |
| medications | | | |
| % Prescribed medication | | | |
| ACEi (%) | | | |
| ARB (%) | | | |
| CCB (%) | | | |
| Alpha-blocker (%) | | | |
| Beta-blocker (%) | | | |
| Diuretics (%) | | | |
| Spironolactone (%) | | | |

Abbreviations: BMI, body mass index; BP, blood pressure; SBP, systolic BP; DBP, diastolic BP; MAP, mean arterial pressure; HR, heart rate; ABPM, ambulatory blood pressure monitoring; MRI, magnetic resonance imaging; TPR, total peripheral resistance; ACEi, angiotensin converting enzyme inhibitor; ARB, angiotensin receptor blocker; CCB, calcium channel blocker. Data are mean ± SD unless otherwise stated. *Fishers exact test.

without VAH at −40 mmHg. There was no decrease in MAP at −20 or −40 mmHg in the group with VAH (*vs.* rest at 0 mmHg). The cardiac index decreased during LBNP at −20 mmHg and −40 mmHg in both groups, with no difference between the groups. Mean changes in MAP, HR and CI showed similar patterns (see Appendix Table A2). The TPR increased during LBNP and there was a group effect on this response. There was an increase in TPR from rest to −20 mmHg in the group with VAH. There was no increase in TPR at −20 mmHg but there was at −40 mmHg in the group with no VAH. When expressed as change in TPR, there was a greater increase in TPR in the group with VAH at −20 mmHg and −40 mmHg *versus* that in the group with no VAH (Δ4.1 ± 2.5 *vs.* Δ1.2 ± 4 mmHg/ml/min; *P* = 0.013 at –20 mmHg and Δ7.1 ± 3.6 *vs.* Δ3.2 ± 3.2 mmHg/ml/min; at −40 mmHg, Appendix Table A2 for mean ± SD and statistical comparisons).

## Cerebral blood flow responses to LBNP

**Basilar artery.** Fig. 1 shows the BA response to LBNP (Appendix Table A3 for mean ± SD). There was an interaction effect on BA blood flow. Interestingly, BA blood flow was not altered during LBNP in patients with VAH but was reduced in patients without VAH at −20 mmHg and −40 mmHg compared to rest. BA blood flow was consistently lower in patients with VAH *versus* patients without VAH, regardless of LBNP level. The change in BA blood flow was greater in the group without VAH *vs.* the group with VAH at −20 mmHg (Δ−8.0 ± 11.7 ml/min *vs.* Δ+5.5 ± 7.3 ml/min and −40 mmHg (Δ−13.6 ± 12.2 ml/min *vs.* Δ−5.4 ± 3.9 ml/min; Fig. 1*B* and Appendix Table A3 for mean data and all statistical comparisons).

There was an interaction effect between LBNP and group on BA conductance (Fig. 1*C*). In both groups,

the BA conductance did not change during LBNP *versus* that at rest (Appendix Table A3 for all mean and ANOVA comparisons). However, BA conductance was lower in patients without VAH at each level of LBNP *vs.* that in the patients with VAH. When expressed as the change in BA conductance from rest to −20 mmHg or −40 mmHg there was an interaction effect between LBNP and group. There was a greater decrease in BA conductance at −20 mmHg in the group without VAH

*versus* those with VAH ($\Delta -0.069 \pm 0.130$ ml/min/mmHg *vs.* $\Delta + 0.041 \pm 0.044$ ml/min/mmHg; Fig. 1*D* and Appendix Table A3 for all ANOVA comparisons).

**Internal carotid artery.** LBNP reduced right and left ICA blood flow in both groups (Fig. 2*A* and *B*). There was no group or interaction effect, suggesting that the decrease in ICA flow in response to LBNP was similar

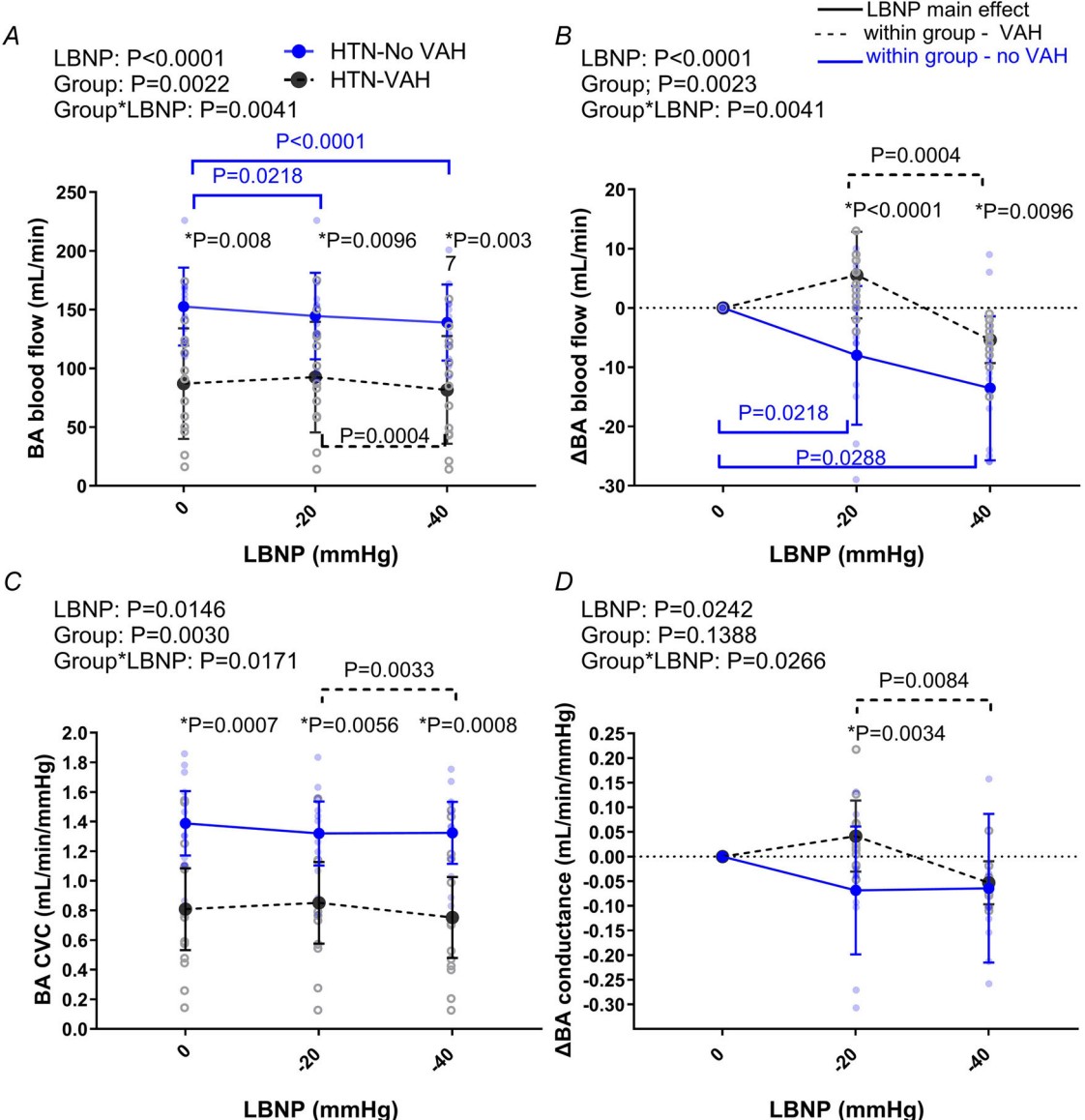

**Figure 1. Basilar artery blood flow and conductance responses to LBNP**
*A* and *C*, basilar artery (BA) blood flow and BA vascular conductance (CVC), respectively, during lower body negative pressure (LBNP) in hypertensive patients with vertebral artery hypoplasia (VAH, $n = 13$) and without VAH (No VAH, $n = 11$). Panels *B* and *D* are the change in BA flow and conductance, respectively. Data compared via two-way mixed model ANOVA. *Between group comparisons with a Bonferroni correction. For all pairwise comparisons and exact *P* values see Appendix Table A3. Blue and black dotted statistic bars above the plots show the specific effect of LBNP in the no VAH group and the VAH group, respectively (Appendix Table A3 for mean data). [Colour figure can be viewed at wileyonlinelibrary.com]

between groups. Right and left ICA conductance were reduced during LBNP (Fig. 2*C* and *D*). The reduction in conductance was not different between groups. Mean changes in ICA flow and conductance are in Appendix Fig. A1.

**Total cerebral blood flow.** An interaction effect was observed for total CBF (Fig. 3; group × LBNP interaction, Appendix Table A5 for mean ± SD). Total CBF did not change with LBNP from rest in patients with VAH; whereas patients without VAH had a reduction in total flow at −20 mmHg and −40 mmHg. When total flow was expressed as a change from rest, there was an interaction effect between group and LBNP, where the reduction in total CBF was larger in the group without

VAH *vs.* those with VAH (Δ−74 ml/min *vs.* −24 ml/min) at −40 mmHg. Total cerebrovascular conductance was reduced by LBNP, but there was no group or interaction effect, suggesting that the total cerebrovascular conductance response to LBNP was similar between groups.

There was an interaction effect on BA blood flow expressed as a percentage of cardiac output. Patients with VAH had an increase in the percentage of cardiac output going to the BA at −20 mmHg and −40 mmHg, whereas patients without VAH had no change in the percentage of cardiac output going to the BA during LBNP. Total CBF as a percentage of cardiac output increased with LBNP. The effect was only seen at −40 mmHg when data for both groups were combined. There was no interaction effect, suggesting that the increase in percentage of

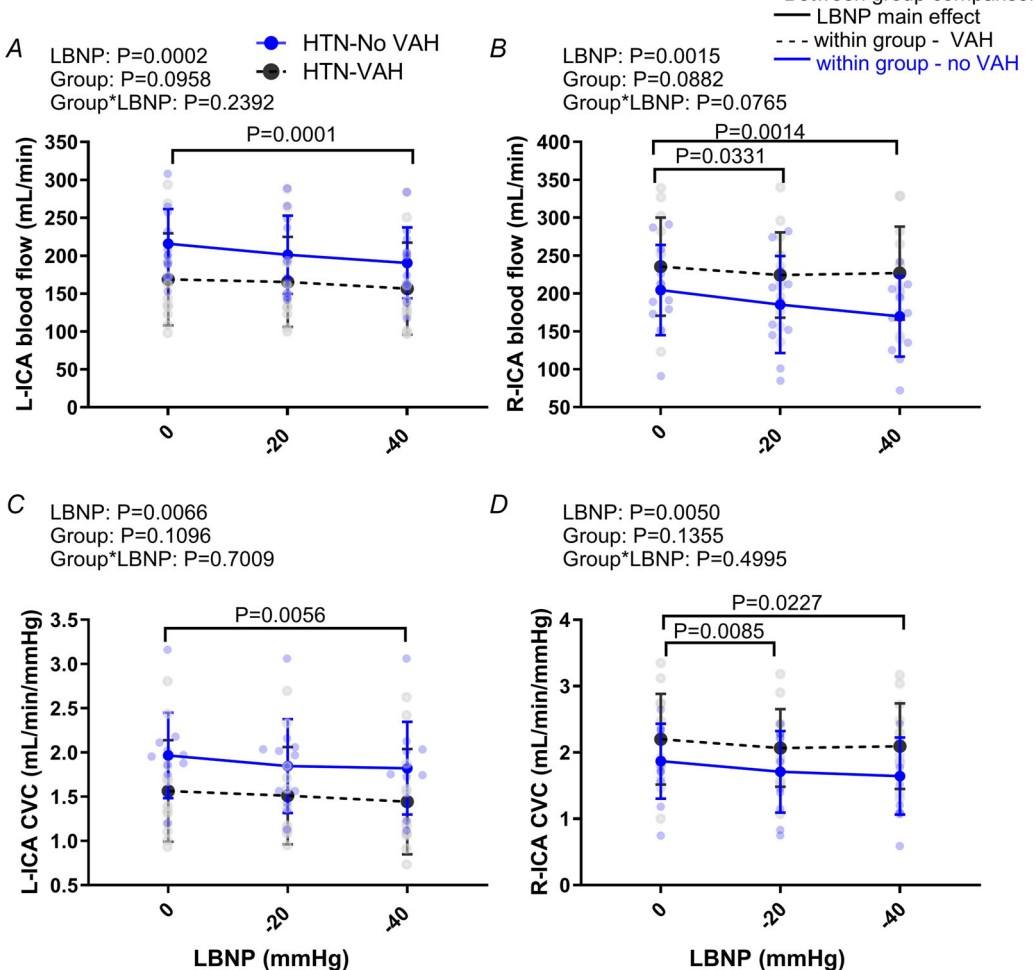

**Figure 2. Internal carotid artery blood flow and conductance responses to LBNP**
Panels *A* and *B* show, respectively, left (L) and right (R) internal carotid artery (ICA) blood flow during lower body negative pressure (LBNP). Panels *C* and *D* show, respectively, L-ICA and R-ICA vascular conductance during (LBNP) in hypertensive (HTN) patients with vertebral artery hypoplasia (VAH, n = 12) and without VAH (No VAH, n = 11). For change in flow or conductance in the L-ICA and R-ICA see Appendix Fig. A1. The black statistic bars above the plots show main effect of LBNP. See Appendix Table A4 for mean ± SD. [Colour figure can be viewed at wileyonlinelibrary.com]

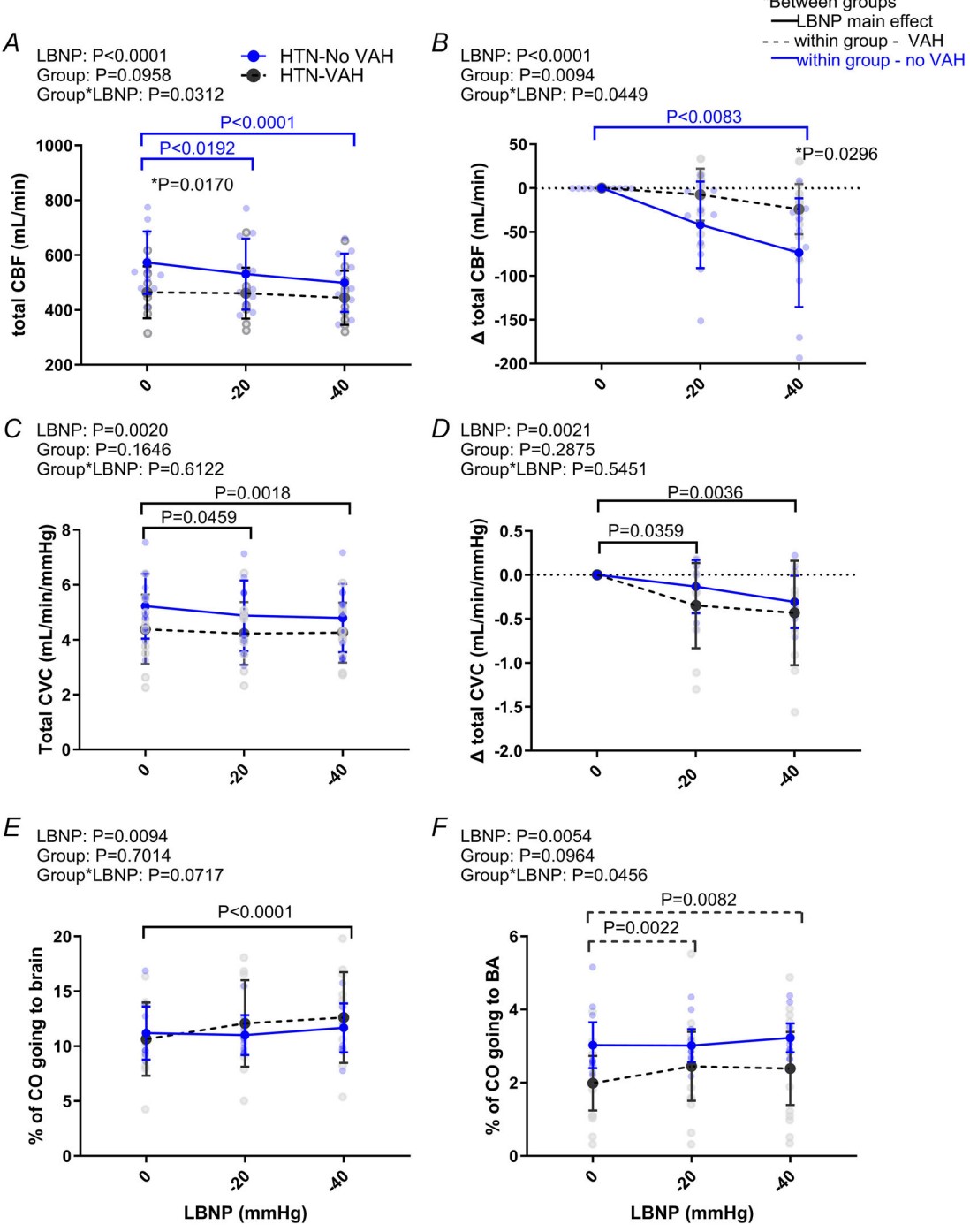

**Figure 3. Total cerebral blood flow and conductance responses to LBNP**

*A* and *C* show, respectively, total cerebral blood flow (CBF) and cerebrovascular conductance (CVC) during lower body negative pressure (LBNP) in hypertensive (HTN) patients with vertebral artery hypoplasia (VAH, *n* = 12) and without VAH (no VAH, *n* = 11). Panels *B* and *D* show the change in total blood flow and conductance. Panels *E* and *F* show total CBF and basilar artery (BA) blood flow as a percentage of the cardiac output (CO). Data compared via two-way mixed model ANOVA. *Between group comparisons with a Bonferroni correction. Blue and black dotted statistic bars above the plots show the specific effect of LBNP in the no VAH group and the VAH group, respectively. The black continuous statistic bars above the plots shows main effect of LBNP. [Colour figure can be viewed at wileyonlinelibrary.com]

cardiac output going to the brain did not depend on the group.

**Partial pressure of end tidal CO₂.** There was a significant effect of LBNP on $P_{ET,CO_2}$ ($P = 0.011$, Fig. 5). The $P_{ET,CO_2}$ did not change from rest to $-20$ mmHg ($P = 0.7106$) but was reduced at $-40$ mmHg *vs.* rest ($P = 0.009$). There was no group ($P = 0.940$) or interaction ($P = 0.369$) effect on $P_{ET,CO_2}$ indicating the decrease in $P_{ET,CO_2}$ with LBNP was similar between groups. When the groups were combined, the change in $P_{ET,CO_2}$ was not correlated to changes total CBF at $-20$ mmHg ($r = 0.27$, $P = 0.307$) or at $-40$ mmHg ($r = -0.27$, $P = 0.306$) LBNP.

## Discussion

Surprisingly, and contrary to the original hypothesis, BA blood flow was unchanged at low levels of LBNP in hypertensive patients with VAH, whereas blood flow decreased in the BA in hypertensive patients without VAH. These changes paralleled the BA vascular conductance response to LBNP, where BA conductance was similar to rest during LBNP in patients with VAH but decreased in patients without VAH. In addition, we report that: (1) the reduction in ICA blood flow in response to LBNP was similar between groups, (2) total CBF was maintained in patients with VAH during LBNP, but decreased during

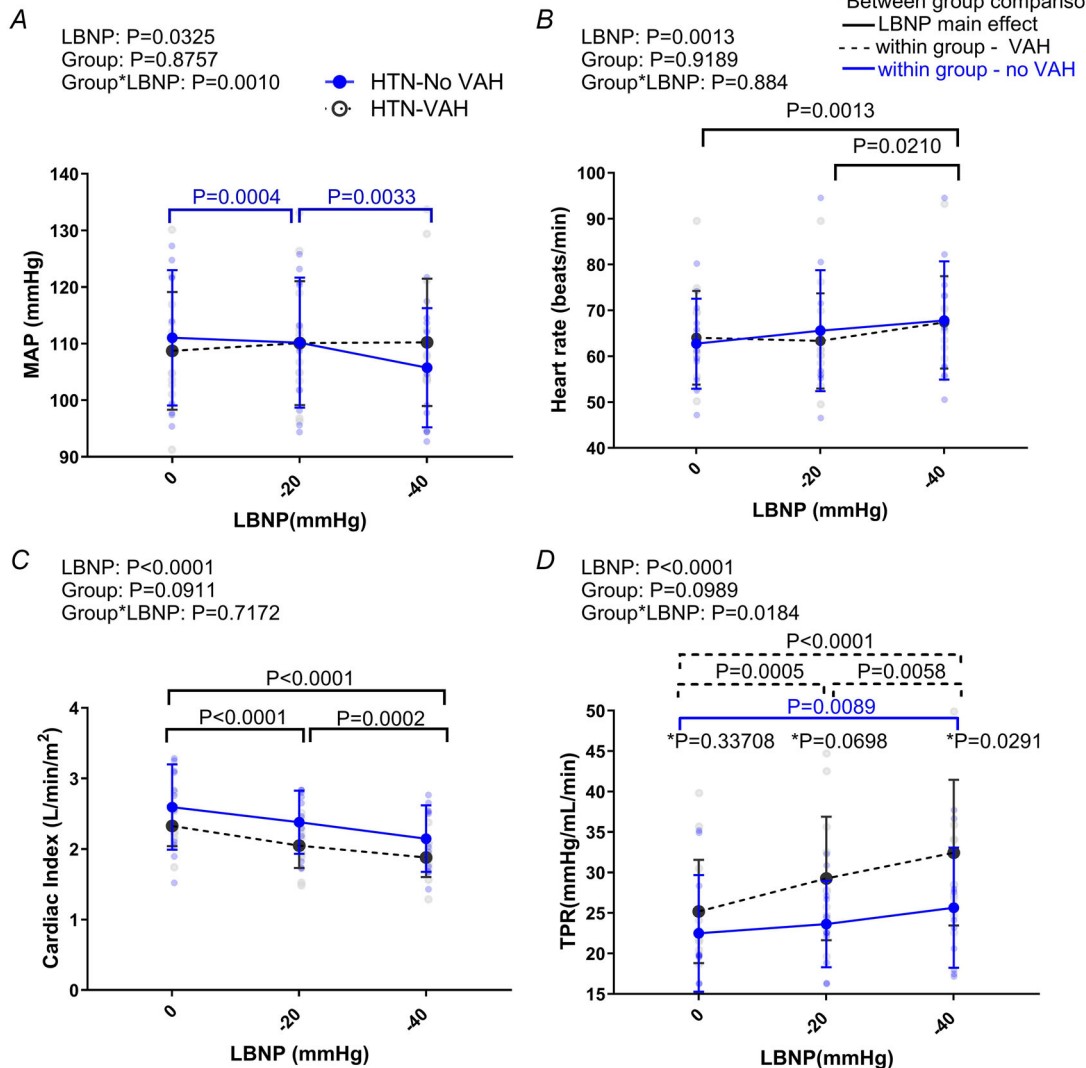

**Figure 4. Systemic haemodynamic responses to LBNP**
Systemic responses to lower body negative pressure (LBNP) in hypertensive (HTN) patients with vertebral artery hypoplasia (VAH, *n* = 13) and without VAH (no VAH, *n* = 11). *A*, mean arterial pressure (MAP); *B*, heart rate; *C*, cardiac index; *D*, total peripheral resistance (TPR). Data compared via two-way mixed model ANOVA. *Between group comparisons with a Bonferroni correction. Blue and black dotted statistic bars above the plots show the specific effect of LBNP in the no VAH group and the VAH group, respectively. The black statistic bars above the plots show main effect of LBNP. [Colour figure can be viewed at wileyonlinelibrary.com]

LBNP in patients with normal anatomy, and (3) interestingly, there was a larger increase in TPR during simulated hypovolaemia in patients with VAH compared to patients with normal calibre vertebral arteries. This greater TPR response probably helped maintain the MAP during LBNP in the VAH group *vs.* the group without VAH where MAP decreased at -40 mmHg.

Taken together, these data suggest that during a reduction in cardiac output, the BA blood flow and total CBF are impacted differently in hypertensive patients with VAH *versus* hypertensive patients with normal vertebral anatomy. Because group differences in the blood flow response to LBNP were not seen in the ICA's, this suggests that VAH has a specific impact on blood flow responses to

hypovolaemia in the BA. This is expected because the VAs feed the BA which then supplies brainstem and cerebellar territories. Additionally, patients with VAH have a larger increase in TPR during LBNP, which may reflect a greater sympathoexcitatory response to simulated hypovolaemia in these patients. This could indicate that when CBF is threatened by a reduction in cardiac output and MAP, individuals with lower basal CBF (such as patients with VAH) have a greater systemic vasoconstrictor response to maintain MAP and CBF (Warnert, Rodrigues et al., 2016). This further indicates that hypertensive patients with VAH may rely on systemic adjustments to maintain CBF and oxygenation (Warnert, Rodrigues et al., 2016).

### Vertebral artery hypoplasia and resting cerebral blood flow

The prevalence of VAH in this small cohort of patients with hypertension (54%) was similar to that reported in our previous study (57%) (Warnert, Rodrigues et al., 2016). Additionally, the hypertensive patients with VAH showed lower blood flow in their BA at rest *versus* patients with normal vertebral anatomy, whereas blood flow was similar in the ICA between groups. This reduction in BA blood flow contributed to lower total CBF in patients with VAH compared to patients with normal calibre vertebral arteries, which is consistent with our previous findings (Warnert, Rodrigues et al., 2016).

### Cerebral blood flow during hypovolaemia

Surprisingly, we found that BA blood flow decreased in patients without VAH during LBNP, whereas BA blood flow was maintained in the patients with VAH. The reduction in BA blood flow in patients without VAH occurred alongside a decrease in BA vascular conductance at −40 mmHg, indicating potential vasoconstriction or a passive change in the volume of the vessel (Whittaker et al., 2017). Other studies have indicated potential vasoconstriction in the MCA (Levine et al., 1994; Sugawara et al., 2017) or vertebral arteries during LBNP (Lewis et al., 2015) in healthy people. This potential vasoconstrictor response is hypothesised to be caused by the dramatic increases in sympathetic nerve activity that occurs during LBNP (Kaur et al., 2018; Lewis et al., 2014; Warnert, Hart et al., 2016). Why this did not happen in the patients with VAH is unclear but may be associated with maintained MAP during LBNP at −40 mmHg in that group whereas MAP decreased in the group without VAH.

Finally, it should be highlighted that, despite no change in BA flow in patients with VAH and a reduction in BA flow in patients without VAH during LBNP, absolute BA

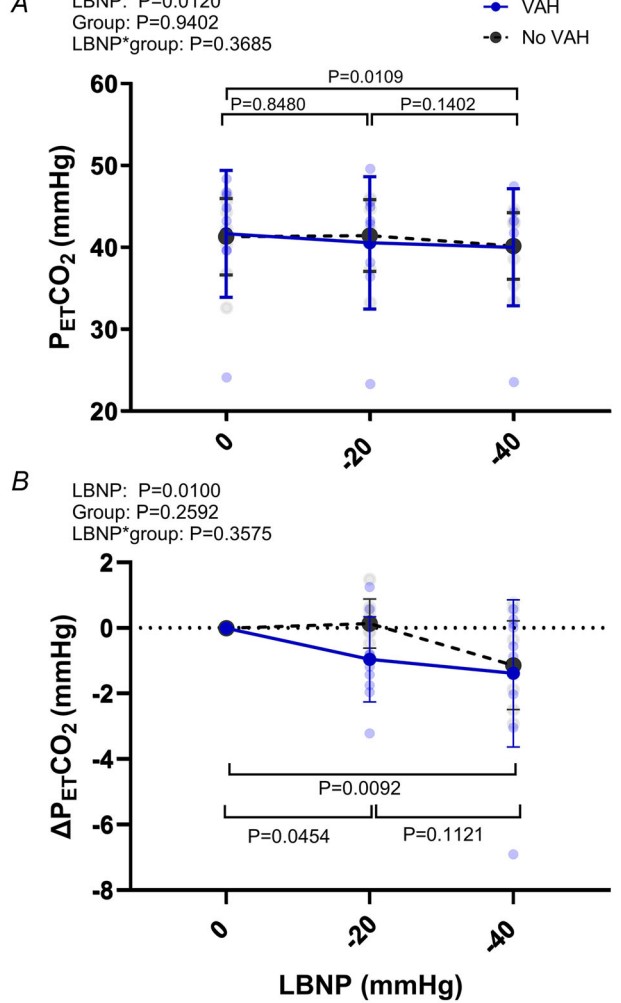

**Figure 5. The end tidal CO2 response to LBNP**
*A* and *B* show, respectively, the partial pressure of the end-tidal $CO_2$ ($P_{ET,CO_2}$) and the change in $P_{ET,CO_2}$ in hypertensive patients with vertebral artery hypoplasia (VAH, $n = 11$) and without VAH (No VAH, $n = 13$) during lower body negative pressure (LBNP). Overall there was a decrease in $P_{ET,CO_2}$ at -40 mmHg. [Colour figure can be viewed at wileyonlinelibrary.com]

blood flow remained lower at every level of LBNP in patients with VAH *versus* patients without VAH.

## Possible mechanisms underlying the unchanged basilar blood flow in patients with VAH

It is possible that there was no change in BA blood flow during LBNP because patients with VAH already have lower resting basilar and global CBF *versus* patients without VAH and could not tolerate further reductions. The overall aim of the cerebral circulation is to maintain adequate blood flow to meet metabolic demand. The normal global cerebral oxygen extraction fraction is 30%–40% (and may vary by region) (Henriksen et al., 2021; Raichle et al., 2001). Therefore, a reduction in CBF is usually met with increased oxygen extraction fraction (Lewis et al., 2014; Powers et al., 1984). Exactly what happens to glucose uptake during reduced blood flow is unclear. Alongside this, recent studies show that in healthy people there is a more than adequate blood supply at rest to meet the resting cerebral metabolic rate of oxygen ($CMRO_2$) (Mergenthaler et al., 2013). Also, when $CMRO_2$ is increased, for example during a cognitive task, the CBF response overshoots the $CMRO_2$ in healthy people. Thus, based on meeting $CMRO_2$, most people can tolerate small reductions in CBF during hypovolaemia.

It is possible that because patients with VAH already have lower CBF, they operate at higher oxygen extraction fractions, in line with what has been previously reported in patients with hypertension (Fujii et al., 1990). Thus, with lower CBF at rest, or in response to an acute reduction in cardiac output, a systemic response is triggered to maintain blood flow in the posterior circulation. Along these lines, the larger increase in TPR in the patients with VAH during LBNP might represent a greater pressor response to the same decrease in cardiac output in these patients and may be a mechanism to help maintain flow in the posterior brain circulation.

### Methodological considerations

The cerebral vessels are sensitive to changes in the partial pressure of arterial $CO_2$ ($P_{aCO_2}$), were a fall in $P_{aCO_2}$ causes the cerebral vessels constrict. We estimated changes in arterial $CO_2$ using $P_{ET,CO_2}$. $P_{ET,CO_2}$ was not reduced at −20 mmHg (Δ−0.57 mmHg) but did decrease at −40 mmHg (Δ−1.5 mmHg). Since $P_{ET}CO_2$ did not change at −20 mmHg, and there was no difference in

the response of $P_{ET,CO_2}$ between groups, it is unlikely that changes in $P_{aCO_2}$ at this stage caused the reduction in CBF in the group without VAH. Therefore, the different response of blood flow in the basilar arteries between groups may not be due to changes in $CO_2$. Finally, the lower sample size may provide the possibility of type II errors (higher possibility of false negatives), given that we have several *P* values that are >0.05 but $P < 0.1$, including that for cardiac index and comparison of changes in flow and conductance in the ICA during LBNP.

## Conclusion

In summary, VAH in patients with hypertension impacts the CBF response to reductions in cardiac output evoked by LBNP. In hypertensive people without VAH, there was a fall in total cerebral and BA blood flow; however, in patients with VAH, total cerebral and BA blood flow was unchanged during moderate levels of LBNP. Interestingly, patients with VAH had a larger rise in TPR during LBNP *versus* the group without VAH. Taken together, we suggest that this indicates (1) patients with hypertension and no VAH can tolerate a reduction in MAP and CBF, without causing an excessive rise in TPR, potentially because they have adequate oxygen extraction reserve and (2) reduced cardiac output triggers a Cushing like response (lower perfusion triggering increased SBP) in patients with hypertension and VAH, so that when CBF is threatened in a circulation with poor reserve (lower basal cerebral perfusion), there is a larger rise in TPR and increased percentage of cardiac output distributed to the BA.

### Perspectives

Hypertensive patients with VAH may rely on their systemic pressor responses to try and maintain perfusion of posterior brain territories in the face of reduced cardiac output. Thus, patients with VAH might need to be treated differently to patients without VAH during situations where cardiac output and the cerebral circulation is threatened – such as during cardiac surgery where risk of poor neurocognitive outcomes is higher (*versus* non-cardiac surgery) (Greaves et al., 2020; Newman et al., 2001). For example, a slightly higher perfusion pressure might be needed in these patients to prevent neurocognitive decline after surgery. This, however, remains to be determined.

## Appendix

Table A1–A5

**Table A1. Pearsons' correlation coefficients and *P* values of body mass index (BMI) *versus* total cerebral blood flow (tCBF) and blood flow in the basilar artery (BA).**

| BMI *vs.* | No VAH (*n* = 11) | VAH (*n* = 13) | Group (*n* = 24) |
|---|---|---|---|
| tCBF at rest *versus* BMI (Pearsons *r*, *P* value) | −0.35, *P* = 0.1222 | −0.44, *P* = 0.1961 | −0.14 *P* = 0.5337 |
| %△tCBF at −20 mmHg | −0.35, *P* = 0.2945 | 0.02, *P* = 0.9625 | −0.34, *P* = 0.1168 |
| %△tCBF at −40 mmHg | −0.44, *P* = 0.1741 | 0.08, *P* = 0.8045 | −0.41, *P* = 0.0830 |
| BA blood flow at rest | −0.23, *P* = 0.5019 | −0.25, *P* = 0.4373 | 0.11, P-0.6280 |
| %△BA blood flow at -20 mmHg | 0.22, *P* = 0.5177 | −0.09, *P* = 0.7906 | −0.19, *P* = 0.3853 |
| %△ BA blood flow at −40 mmHg | 0.03, *P* = 0.9318 | 0.09, *P* = 0.7826 | 0.04, *P* = 0.8690 |

**Table A2. Systemic cardiovascular responses during lower body negative pressure (LBNP)**

| | No VAH (*n* = 11) | VAH (*n* = 13) | All participants (*n* = 24) | Pairwise comparison (*P* value) | ANOVA |
|---|---|---|---|---|---|
| MAP | 111 ± 12 | 109 ± 10 | 110 ± 11 | *P* = 0.6146 | *LBNP: P* = 0.0325 |
| Mean ± SD (mmHg) | 110 ± 12 | 110 ± 11 | 110 ± 11 | *P* = 0.9859 | *Group: P* = 0.8757 |
| 0 mmHg | 106 ± 11 | 110 ± 11 | 108 ± 11 | *P* = 0.3273 | *Group*LBNP:P* = 0.0010 |
| −20 mmHg | *P* > 0.9999 | *P* = 0.7395 | *P* > 0.9999 | | |
| −40 mmHg | *P* = 0.0004 | *P* = 0.6093 | *P* = 0.1009 | | |
| *Main effect LBNP (P values)* | *P* = 0.0033 | *P* > 0.9999 | *P* = 0.0493 | | |
| 0 *vs.* −20 mmHg | | | | | |
| 0 *vs.* −40 mmHg | | | | | |
| −20 *vs.* −40 mmHg | | | | | |
| △MAP | −0.9 ± 2.3 | 1.4 ± 2.3 | 0.4 ± 2.5 | 0.3430 | *LBNP: P* = 0.0382 |
| Mean ± SD (mmHg) | −5.3 ± 6.0 | 1.5 ± 1.1 | −1.6 ± 6.0 | 0.0002 | *Group: P* = 0.0018 |
| −20 mmHg | 0.0107 | *P* > 0.9999 | | | *Group*LBNP: P* = 0.0284 |
| −40 mmHg | | | | | |
| *Main effect LBNP (P values)* | | | | | |
| −20 *vs.* −40 mmHg | | | | | |
| Heart rate | 63 ± 10 | 64 ± 10 | 63 ± 10 | | *LBNP: P* = 0.0013 |
| Mean ± SD (beats/min) | 66 ± 13 | 63 ± 10 | 64 ± 12 | | *Group: P* = 0.9189 |
| −0 mmHg | 68 ± 13 | 67 ± 10 | 68 ± 11 | | *Group*LBNP:P* = 0.2884 |
| −20 mmHg | | | *P* > 0.9999 | | |
| −40 mmHg | | | *P* = 0.0013 | | |
| *Main effect LBNP (P values)* | | | *P* = 0.0210 | | |
| 0 *vs.* −20 mmHg | | | | | |
| 0 *vs.* −40 mmHg | | | | | |
| −20 *vs.* −40 mmHg | | | | | |

*(Continued)*

**Table A2. (Continued)**

| | No VAH (*n* = 11) | VAH (*n* = 13) | All participants (*n* = 24) | Pairwise comparison (*P* value) | ANOVA |
|---|---|---|---|---|---|
| △Heart rate | 3 ± 9 | −1 ± 2 | 1 ± 7 | | *LBNP: P = 0.1547* |
| Mean ± SD (mmHg) | 2 ± 3 | 4 ± 3 | 3 ± 3 | | *Group: P = 0.5516* |
| −20 mmHg | | | | | *Group\*LBNP:P = 0.0644* |
| −40 mmHg | | | | | |
| Cardiac index | 2.6 ± 0.6 | 2.3 ± 0.3 | 2.5 ± 0.4 | | *LBNP: P < 0.0001* |
| Mean ± SD (L/min/m$^2$) | 2.4 ± 0.5 | 2.1 ± 0.3 | 2.2 ± 0.4 | | *Group: P = 0.0911* |
| 0 mmHg | 2.2 ± 0.5 | 1.9 ± 0.3 | 2.0 ± 0.4 | | *Group\*LBNP: P = 0.7172* |
| −20 mmHg | | | *P* < 0.0001 | | |
| −40 mmHg | | | *P* < 0.0001 | | |
| *Main effect LBNP (P values)* | | | *P* = 0.0002 | | |
| 0 *vs.* −20 mmHg | | | | | |
| 0 *vs.* −40 mmHg | | | | | |
| −20 *vs.* −40 mmHg | | | | | |
| △Cardiac index | −0.4 ± 0.6 | −0.5 ± 0.3 | −0.5±0.5 | | *LBNP: P < 0.0001* |
| (l/min) | −0.9 ± 0.6 | −0.9 ± 0.3 | −0.9±0.5 | | *Group: P = 0.9143* |
| −20 mmHg | | | <0.0001 | | *Group\*LBNP: P = 0.2734* |
| −40 mmHg | | | | | |
| *Main effect LBNP (P values)* | | | | | |
| −20 *vs.* -40 mmHg | | | | | |
| Cardiac output | 5.3 ± 1.3 | 4.5 ± 0.8 | 4.9 ± 1.1 | | *LBNP: P < 0.0001* |
| (l/min) | 4.9 ± 1.0 | 3.9 ± 0.8 | 4.4 ± 1.0 | | *Group: P = 0.0502* |
| Mean ± SD | 4.4 ± 1.1 | 3.6 ± 0.8 | 4.0 ± 1.0 | | *Group\*LBNP: P = 0.7072* |
| −0 mmHg | | | | | |
| −20 mmHg | | | | | |
| −40 mmHg | | | | | |
| TPR (mmHg/ml/min) | 22.5 ± 7.2 | 25.2 ± 6.4 | 23.8 ± 6.3 | *P* = 0.3708 | *LBNP: P < 0.0001* |
| Mean ± SD | 23.6 ± 5.4 | 29.3 ± 7.6 | 26.5 ± 6.3 | *P* = 0.0698 | *Group: P = 0.0989* |
| −0 mmHg | 25.7 ± 7.4 | 32.5 ± 9.0 | 29.1 ± 7.8 | *P* = 0.0291 | *Group\*LBNP: P = 0.0184* |
| −20 mmHg | *P* = 0.7734 | *P* = 0.0005 | | | |
| −40 mmHg | *P* = 0.0089 | *P* < 0.0001 | | | |
| *Main effect LBNP (P values)* | *P* = 0.1539 | *P* = 0.0058 | | | |
| 0 *vs.* −20 mmHg | | | | | |
| 0 *vs.* −40 mmHg | | | | | |
| −20 *vs.* −40 mmHg | | | | | |
| △TPR | 1.2 ± 4 | 4.1 ± 2.5 | 2.6 ± 3.5 | *P* = 0.0129 | *LBNP: P < 0.0001* |
| (mmHg/ml/min) | 3.2 ± 3 | 7.1 ± 3.6 | 5.1 ± 4.1 | *P* = 0.0016 | *Group: P = 0.0112* |
| Mean ± SD | *P* = 0.1599 | *P* = 0.0157 | *P* = 0.0035 | | *Group\*LBNP: P = 0.0248* |
| −20 mmHg | | | | | |
| −40 mmHg | | | | | |
| *Main effect LBNP (P values)* | | | | | |
| −20 *vs.* −40 mmHg | | | | | |

Abbreviations: MAP, mean arterial pressure; TPR, total peripheral resistance. Data are mean ± standard deviation along with two-way mixed model ANOVA data. *P* values for the effect of LBNP are shown within each group only if there was a main interaction effect between the group and LBNP. The *P* value for the effect of LBNP in all participants grouped together is shown if there was only a main effect of LBNP on the MAP, heart rate, cardiac output and TPR.

**Table A3. Total basilar artery (BA) blood flow and BA vascular conductance during lower body negative pressure (LBNP)**

| | No VAH (n = 11) | VAH (n = 13) | All participants (n = 24) | Pairwise comparison (P value) | ANOVA |
|---|---|---|---|---|---|
| Total BA flow | 153 ± 33 | 87 ± 47 | 120 ± 46 | P = 0.0003 | *LBNP: P < 0.0001* |
| Mean ± SD (ml/min) | 145 ± 37 | 93 ± 47 | 119 ± 37 | P = 0.0032 | *Group: P = 0.0022* |
| 0 mmHg | 139 ± 32 | 82 ± 46 | 110 ± 41 | P = 0.0012 | *Group*LBNP: P = 0.0041* |
| −20 mmHg | P = 0.0218 | P = 0.1195 | | | |
| −40 mmHg | P = 0.0001 | P = 0.1362 | | | |
| *Main effect LBNP (P values)* | P = 0.1723 | P = 0.0004 | | | |
| 0 *vs.* −20 mmHg | | | | | |
| 0 *vs.* −40 mmHg | | | | | |
| −20 *vs.* −40 mmHg | | | | | |
| ΔBA flow | −8.0 ± 11.7 | 5.5 ± 7.3 | −1.2 ± 9.4 | P < 0.0001 | *LBNP: P < 0.0001* |
| Mean ± SD (ml/min) | −13.6 ± 12.2 | −5.4 ± 3.9 | −9.5 ± 7.9 | P = 0.0096 | *Group: P = 0.0023* |
| −20 mmHg | P = 0.0218 | P = 0.1195 | P = >0.9999 | | *Group*LBNP: P = 0.0041* |
| −40 mmHg | P < 0.0001 | P = 0.1362 | P < 0.0001 | | |
| *Main effect LBNP (P values)* | P = 0.1720 | P = 0.0004 | P = 0.0003 | | |
| 0 *vs.* −20 mmHg | | | | | |
| 0 *vs.* −40 mmHg | | | | | |
| −20 *vs.* −40 mmHg | | | | | |
| Total BA conductance | 1.4 ± 0.3 | 0.8 ± 0.5 | 1.1 ± 0.5 | P = 0.0007 | *LBNP: P = 0.0133* |
| Mean ± SD | 1.3 ± 0.3 | 0.9 ± 0.5 | 1.1 ± 0.4 | P = 0.0053 | *Group: P = 0.0029* |
| (ml/min/mmHg) | 1.3 ± 0.3 | 0.8 ± 0.5 | 1.0 ± 0.4 | P = 0.0007 | *Group*LBNP: P = 0.0164* |
| 0 mmHg | P = 0.0932 | P = 0.4322 | P > 0.9999 | | |
| −20 mmHg | P = 0.1287 | P = 0.1357 | P = 0.0162 | | |
| −40 mmHg | P > 0.9999 | P = 0.0033 | P = 0.0687 | | |
| *Main effect LBNP (P values)* | | | | | |
| 0 *vs.* −20 mmHg | | | | | |
| 0 *vs.* −40 mmHg | | | | | |
| −20 *vs.* −40 mmHg | | | | | |
| ΔBA conductance | −0.07 ± 0.13 | 0.04 ± 0.07 | −0.01 ± 0.10 | P = 0.0036 | *LBNP: P = 0.0213* |
| Mean ± SD (%) | −0.06 ± 0.15 | −0.06 ± 0.04 | −0.06 ± 0.10 | P = 0.8580 | *Group: P = 0.1575* |
| −20 mmHg | P = 0.1006 | P = 0.5210 | P > 0.9999 | | *Group*LBNP: P = 0.0249* |
| −40 mmHg | P = 0.1380 | P = 0.2037 | P = 0.0244 | | |
| *Main effect LBNP (P values)* | P = 0.9999 | P = 0.0075 | P = 0.1102 | | |
| 0 *vs.* −20 mmHg | | | | | |
| 0 *vs.* −40 mmHg | | | | | |
| −20 *vs.* −40 mmHg | | | | | |

Data are mean ± standard deviation along with two-way mixed model ANOVA data. *P* values for the effect of LBNP are shown within each group only if there was a main interaction effect between the group and LBNP. The *P* value for the effect of LBNP in all participants grouped together is shown if there was only a main effect of LBNP on BA blood flow and conductance.

**Table A4. Left (L) and right (R) internal carotid artery (ICA) blood flow and ICA vascular conductance during lower body negative pressure (LBNP)**

| | No VAH (n = 11) | VAH (n = 13) | All participants (n = 24) | Pairwise comparison (P value) | ANOVA |
|---|---|---|---|---|---|
| L-ICA flow | 216 ± 46 | 169 ± 61 | 192 ± 58 | | *LBNP: P = 0.0002* |
| Mean ± SD (ml/min) | 201 ± 52 | 165 ± 59 | 183 ± 58 | | *Group: P = 0.0958* |
| 0 mmHg | 191 ± 47 | 156 ± 61 | 174 ± 56 | | *Group*LBNP: P = 0.2392* |
| −20 mmHg | | | *P > 0.1080* | | |
| −40 mmHg | | | *P = 0.0001* | | |
| *Main effect LBNP (P values)* | | | *P = 0.0693* | | |
| 0 *vs.* −20 mmHg | | | | | |
| 0 *vs.* −40 mmHg | | | | | |
| −20 *vs.* −40 mmHg | | | | | |
| R-ICA flow | 204 ± 60 | 235 ± 65 | 220 ± 63 | | *LBNP: P = 0.0015* |
| Mean ± SD (ml/min) | 185 ± 64 | 224 ± 56 | 205 ± 62 | | *Group: P = 0.0882* |
| 0 mmHg | 170 ± 53 | 227 ± 61 | 198 ± 63 | | *Group*LBNP: P = 0.0765* |
| −20 mmHg | | | *P = 0.0331* | | |
| −40 mmHg | | | *P = 0.0014* | | |
| *Main effect LBNP (P values)* | | | *P = 0.7820* | | |
| 0 *vs.* −20 mmHg | | | | | |
| 0 *vs.* −40 mmHg | | | | | |
| −20 *vs.* −40 mmHg | | | | | |
| L-ICA conductance | 1.97 ± 0.48 | 1.63±0.56 | 1.79±0.54 | | *LBNP: P = 0.0047* |
| Mean ± SD | 1.85 ± 0.53 | 1.57±0.55 | 1.70±0.55 | | *Group: P = 0.1738* |
| (ml/min/mmHg) | 1.82 ± 0.52 | 1.49±0.57 | 1.66±0.57 | | *Group*LBNP: P = 0.6991* |
| 0 mmHg | | | *P = 0.1104* | | |
| −20 mmHg | | | *P = 0.0038* | | |
| −40 mmHg | | | *P = 0.5919* | | |
| *Main effect LBNP (P values)* | | | 0.1104 | | |
| 0 *vs.* −20 mmHg | | | 0.0038 | | |
| 0 *vs.* −40 mmHg | | | 0.5919 | | |
| −20 *vs.* −40 mmHg | | | | | |
| R-ICA conductance | 1.87 ± 0.56 | 2.26±0.71 | 2.05±0.66 | | *LBNP: P = 0.0060* |
| Mean ± SD | 1.71 ± 0.61 | 2.11±0.61 | 1.90±0.63 | | *Group: P = 0.1071* |
| 0 mmHg | 1.64 ± 0.58 | 2.14±0.69 | 1.89±0.66 | | *Group*LBNP: P = 0.6313* |
| −20 mmHg | | | *P = 0.0262* | | |
| −40 mmHg | | | *P = 0.0103* | | |
| *Main effect LBNP (P values)* | | | *P > 0.9999* | | |
| 0 *vs.* −20 mmHg | | | | | |
| 0 *vs.* −40 mmHg | | | | | |
| −20 *vs.* −40 mmHg | | | | | |

Data are mean ± standard deviation along with two-way mixed model ANOVA data. P values for the effect of LBNP are shown within each group only if there was a main interaction effect between the group and LBNP. The P value for the effect of LBNP in all participants grouped together is shown if there was only a main effect of LBNP on ICA blood flow and conductance.

**Table A5. Total cerebral blood flow (CBF) and vascular conductance (CVC) during lower body negative pressure (LBNP)**

| | No VAH (n = 11) | VAH (n = 13) | All participants (n = 24) | Pairwise comparison (P value) | ANOVA |
|---|---|---|---|---|---|
| Total CBF | 572 ± 114 | 456 ± 111 | 515 ± 83 | P = 0.0170 | LBNP: P < 0.0001 |
| Mean ± SD (ml/min) | 531 ± 130 | 453 ± 130 | 492 ± 55 | P = 0.1059 | Group: P = 0.0958 |
| 0 mmHg | 499 ± 107 | 436 ± 113 | 468 ± 45 | P = 0.1903 | Group*LBNP: P = 0.0312 |
| −20 mmHg | P = 0.0192 | P > 0.9999 | P = 0.0935 | | |
| −40 mmHg | P < 0.0001 | P = 0.4877 | P = 0.0001 | | |
| Main effect LBNP (P values) | P = 0.1054 | P = 0.7170 | P = 0.0629 | | |
| 0 vs. −20 mmHg | | | | | |
| 0 vs. −40 mmHg | | | | | |
| −20 vs. −40 mmHg | | | | | |
| Δ Total CBF | −42 ± 49 | −7 ± 30 | −25 ± 42 | P = 0.0610 | LBNP: P < 0.0001 |
| Mean ± SD (ml/min) | −74 ± 62 | −24 ± 27 | −49 ± 53 | P = 0.0296 | Group: P = 0.0094 |
| −20 mmHg | 0.0547 | >0.9999 | 0.0431 | | Group*LBNP:P = 0.0449 |
| −40 mmHg | 0.0083 | 0.0436 | 0.0008 | | |
| Main effect LBNP (P values) | 0.4605 | 0.3175 | 0.1157 | | |
| 0 vs. −20 mmHg | | | | | |
| 0 vs. −40 mmHg | | | | | |
| −20 vs. −40 mmHg | | | | | |
| Total CVC | 5.2 ± 1.2 | 4.4 ± 1.2 | 4.9 ± 0.7 | 0.1135 | LBNP: P = 0.0020 |
| Mean ± SD | 4.9 ± 1.3 | 4.2 ± 1.1 | 4.6 ± 0.5 | 0.2229 | Group: P = 0.1646 |
| (ml/min/mmHg) | 4.8 ± 1.2 | 4.1 ± 1.1 | 4.4 ± 0.5 | 0.1718 | Group*LBNP:P = 0.6122 |
| 0 mmHg | 0.0507 | 0.8442 | 0.0459 | | |
| −20 mmHg | 0.0106 | 0.1006 | 0.0018 | | |
| −40 mmHg | >0.9999 | 0.7721 | 0.6555 | | |
| Main effect LBNP (P values) | | | | | |
| 0 vs. −20 mmHg | | | | | |
| 0 vs. −40 mmHg | | | | | |
| −20 vs. −40 mmHg | | | | | |
| Δ Total CVC | −0.35 ± 0.48 | −0.13 ± 0.30 | −0.24 ± 0.41 | | LBNP: P = 0.0021 |
| Mean ± SD | −0.43 ± 0.59 | −0.31 ± 0.30 | −0.37 ± 0.46 | | Group: P = 0.2875 |
| (ml/min/mmHg) | | | 0.0359 | | Group*LBNP: P = 0.5451 |
| −20 mmHg | | | 0.0036 | | |
| −40 mmHg | | | 0.6527 | | |
| Main effect LBNP (P values) | | | | | |
| 0 vs. −20 mmHg | | | | | |
| 0 vs. −40 mmHg | | | | | |
| −20 vs. −40 mmHg | | | | | |
| BA flow as % of CO | 3.0 ± 0.9 | 2.0 ± 1.2 | 2.5 ± 1.0 | 0.0344 | LBNP: P = 0.0054 |
| Mean ± SD (%) | 3.0 ± 0.7 | 2.5 ± 1.5 | 2.7 ± 1.2 | 0.2401 | Group: P = 0.0964 |
| 0 mmHg | 3.2 ± 0.6 | 2.4 ± 1.6 | 2.8 ± 1.3 | 0.0856 | Group*LBNP: P = 0.0456 |
| −20 mmHg | >0.9999 | 0.0022 | | | |
| −40 mmHg | 0.3968 | 0.0082 | | | |
| Main effect LBNP (P values) | 0.3508 | >0.9999 | | | |
| 0 vs. −20 mmHg | | | | | |
| 0 vs. −40 mmHg | | | | | |
| −20 vs. −40 mmHg | | | | | |

(Continued)

**Table A5. (Continued)**

| | No VAH (*n* = 11) | VAH (*n* = 13) | All participants (*n* = 24) | Pairwise comparison (*P* value) | ANOVA |
|---|---|---|---|---|---|
| Total CBF as % of CO | 11 ± 2 | 11 ± 2 | 11 ± 2 | | *LBNP: P = 0.0094* |
| Mean ± SD (%) | 11 ± 2 | 12 ± 4 | 11 ± 3 | | *Group: P = 0.7014* |
| 0 mmHg | 12 ± 2 | 13 ± 4 | 12 ± 3 | | *Group*LBNP: P = 0.0717* |
| −20 mmHg | | | | 0.3202 | |
| −40 mmHg | | | | 0.0072 | |
| Main effect LBNP (*P* values) | | | | 0.3615 | |
| 0 *vs.* −20 mmHg | | | | | |
| 0 *vs.* −40 mmHg | | | | | |
| −20 *vs.* −40 mmHg | | | | | |

Data are mean ± standard deviation along with two-way mixed model ANOVA data. *P* values for the effect of LBNP are shown within each group only if there was a main interaction effect between the group and LBNP. The *P* value for the effect of LBNP in all participants grouped together is shown if there was only a main effect of LBNP on CBF and CVC.

Figure A1

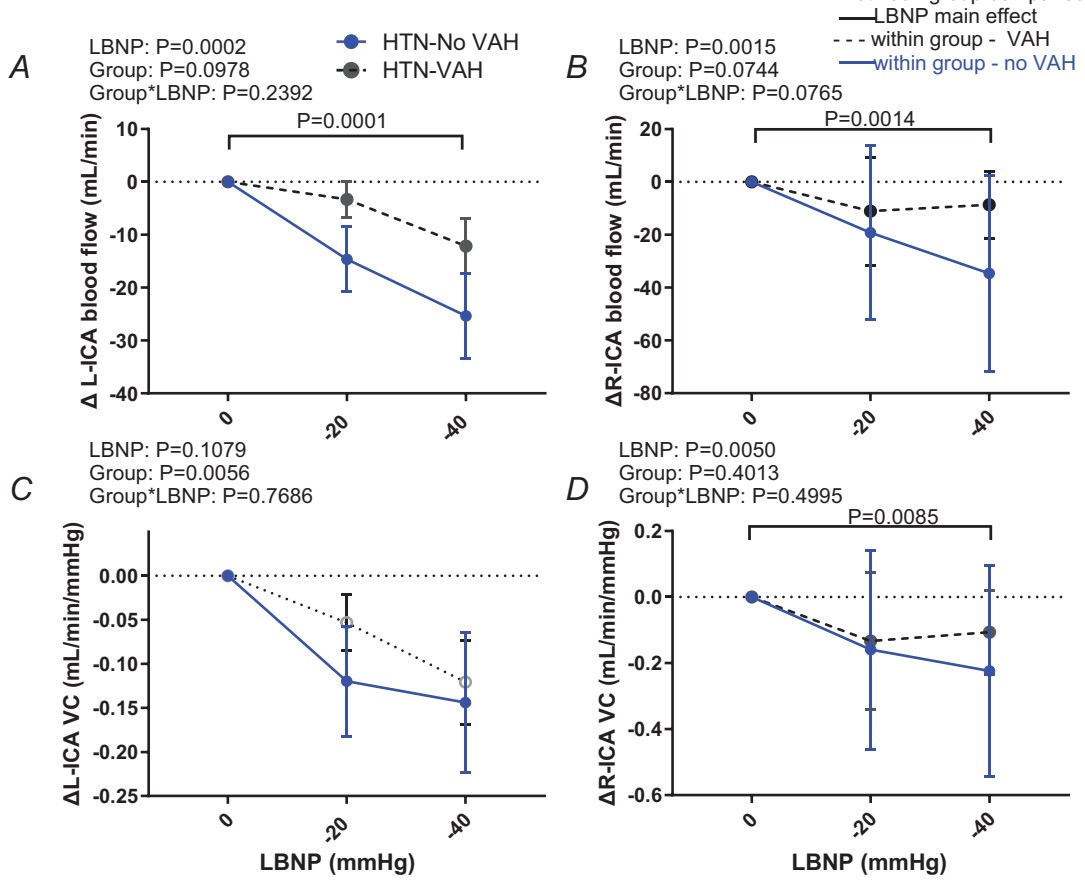

**Figure A1. The change in interbal carotid artery blood flow during LBNP**

*A* and *B*, change in left and right internal carotid artery (L-ICA; R-ICA) blood flow (*A*) and *v*ascular conductance (VC; *B*) in hypertensives with vertebral artery hypoplasia (VAH) and the hypertensives without VAH. [Colour figure can be viewed at wileyonlinelibrary.com]

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

## Additional information

### Data availability statement

Data are available from the authors on reasonable request.

### Competing interests

The views expressed in this manuscript are those of the authors and not necessarily those of the institution or funders.

### Author contributions

S.N. and E.C.H. contributed to the conception and the design of the study. All authors contributed to the acquisition, analysis or interpretation of data for the work and all authors drafting the work or revising it critically for important intellectual content. All authors have approved the final version of the manuscript and agree to be accountable for all aspects of the work in ensuring that questions related to the accuracy or integrity of any part of the work are appropriately investigated and resolved. All persons designated as authors qualify for authorship, and all those who qualify for authorship are listed.

### Funding

This study was supported by the Wellcome Trust PhD programme (Neural Dyanmics) and British Heart Foundation; AA/18/1/34219. The James Tudor Foundation provided financial support for Research Nurse time.

### Acknowledgements

Thank you to (A) the participants for taking part and (B) the BHI Cardiology Research nurses; we would not have been able to complete these studies without them.

### Keywords

cerebral blood flow, hypertension, LBNP, MRI, vertebral artery hypoplasia

## Supporting information

Additional supporting information can be found online in the Supporting Information section at the end of the HTML view of the article. Supporting information files available:

**Peer Review History**

