## [Peer Review History · The Journal of Physiology]

Cerebral blood flow during simulated central hypovolaemia in people with hypertension: does vertebral artery hypoplasia matter?

Sandra Neumann, Jonathan Rodrigues, Lydia L Simpson, Chris B Lawton, Daniel Burden, Matthew D Kobetic, Zoe H. Adams, Katrina Hope, Julian F. R. Paton, Hazel C. Blythe, Nathan E Manghat, Jill N Barnes, Angus K. Nightingale, Mark Hamilton, and Emma C Hart

DOI: 10.1113/JP287786

Corresponding author(s): Emma Hart (Emma.Hart@bristol.ac.uk)

The following individual(s) involved in review of this submission have agreed to reveal their identity: Jay M.J.R. Carr (Referee #1)

Review Timeline:

Submission Date:	07-Oct-2024
Editorial Decision:	23-Oct-2024
Revision Received:	08-Jan-2025
Accepted:	28-Jan-2025

Senior Editor: Harold Schultz

Reviewing Editor: Philip Ainslie

Transaction Report:

Dear Dr Hart,

Re: JP-RP-2024-287786 "Cerebral blood flow during simulated central hypovolaemia in people with hypertension: does vertebral artery hypoplasia matter?" by Sandra Neumann, Jonathan Rodrigues, Lydia L Simpson, Chris B Lawton, Daniel Burden, Matthew D Kobetic, Zoe H. Adams, Katrina Hope, Julian F. R. Paton, Hazel C. Blythe, Nathan E Manghat, Jill N Barnes, Angus K. Nightingale, Mark Hamilton, and Emma C Hart

Thank you for submitting your manuscript to The Journal of Physiology. It has been assessed by a Reviewing Editor and by 2 expert referees and we are pleased to tell you that it is potentially acceptable for publication following satisfactory major revision.

LANGUAGE EDITING AND SUPPORT FOR PUBLICATION: If you would like help with English language editing, or other article preparation support, Wiley Editing Services offers expert help, including English Language Editing, as well as translation, manuscript formatting, and figure formatting at www.wileyauthors.com/eoo/preparation. You can also find resources for Preparing Your Article for general guidance about writing and preparing your manuscript at www.wileyauthors.com/eoo/prepresources.

REVISION CHECKLIST:

We look forward to receiving your revised submission.

Yours sincerely,

Harold Schultz
Senior Editor
The Journal of Physiology

EDITOR COMMENTS

Reviewing Editor: Comments to the Author:

Please see the generally positive reviews from two experts in the field. Although merit in your study is noted, there are some important comments that need to be thoroughly addressed in a revised submission.

Senior Editor:

Comments to the Author:

Thank you for submission of your research article to the Journal of Physiology for consideration. The article has been reviewed by experts in the field and found to be potentially acceptable for publication pending adequate revision to address all of the concerns raised. Please address all comments from the external referees and reviewing editor as well as addressing the list of requirements or publication in the journal described in this letter and rigour and reproducibility requirements as described in the link below.

<https://physoc.onlinelibrary.wiley.com/pb-assets/hub-assets/physoc/documents/TJP-Rigour-and-Reproducibility-Requirements-1724673661727.pdf>

Please note deficiencies in the section on human experiments.

We encourage authors to follow best practices of rigour and reproducibility as defined by the Journal below.

<https://physoc.onlinelibrary.wiley.com/pb-assets/hub-assets/physoc/documents/TJP-Rigour-and-Reproducibility-Best-Practices-and-Resources.pdf>

REFEREE COMMENTS

Referee #1:

See attached

Referee #2:

I appreciate the opportunity to review the manuscript entitled "Cerebral blood flow during simulated central hypovolaemia in people with hypertension: does vertebral artery hypoplasia matter?" by Neumann and colleagues.

The present study aimed to explore whether vertebral artery hypoplasia (VAH) in hypertensive patients impacts cerebral blood flow (CBF) regulation during haemodynamic stress (reduced cardiac output and blood pressure) induced via body negative pressure (LBNP). The authors conclude that when compared to hypertensive patients without VAH, patients with VAH evoke a greater systemic peripheral resistance to ultimately preserve CBF during LBNP-induced hypovolemia. This research holds clinical relevance given the prevalence of ischaemic stroke in patients with VAH and the authors deserve commendation for their work in this patient group. Please see below for my constructive comments to the manuscript.

L134: Abbreviate basilar artery to BA here and then amend throughout the manuscript.

L150: How was your sample size determined and are the authors confident in their ability to detect subtle differences in their variables of interest?

L152: Please include the appropriate reference number for local R&D approval.

L156: Please state whether this study was conducted in accordance with the Declaration of Helsinki.

L165: You have not defined ABPM.

L203: Can the authors confirm the timeframe/order of the cardiac MRI in relation to ICA/BA imaging?

L214: Please include manufacturer details for the LBNP barometer.

L225: Please include manufacturer details for the sphygmomanometer.

L237: Please state total CBF here.

L240: Please abbreviate total peripheral resistance to TPR.

L241: Please include the calculation for cardiac index here and remove from L286.

L241: L/min not l/min. Please correct throughout the manuscript.

L241: Please abbreviate cardiac output to CO.

L243: mL not ml. Please correct throughout the manuscript.

L257: Please provide more detail on "all analyses were blinded".

L273: The results section is rather heavy. Where most variables have been abbreviated, please could you use the abbreviation hence forth to reduce text length.

L273: Within the results section, p-values do not need to be greater than 3 decimal places (as seen in table 1) - please amend where appropriate within the text. The figures are fine having 4 decimal places.

L301: Remove "Bonferroni multiple comparisons". This is stated in the statistical analysis section.

L317: Units for cerebral vascular conductance should be mL/min/mmHg not mL/mmHg/min.

L319: Up until this point the authors have provided absolute values or delta values for variables but have not for the remaining results sections. Either add these to the remaining sections or remove them all together.

L509: The methodological considerations could be written with fewer short sentences. Please remove the values provided in the brackets as it's not clear whether this is a total group observation or a specific group.

L530: It is not clear what a "Cushing like response" is referring to.

L555: Could a funding reference number be added for the Wellcome Trust?

L680: Table 1. Please confirm the p-values provided for the clinic BP and HR and ABPM are correct as they appear to be identical to each other.

END OF COMMENTS

Congratulations on a very well conducted study and well-presented manuscript. I have only a small number of comments. See below, in no particular order.

1. Line 486 – “The normal global cerebral oxygen extraction fraction is 15-20% (and may vary by region), with a potential maximum of 70-80%”. I don’t believe the paper you reference here measured or reported oxygen extraction fraction, so perhaps find a more appropriate paper to cite. Additionally, I’m not sure that these values are entirely correct. To my knowledge, cerebral OEF is usually around 30%, give or take ~5-10%, and I don’t think I have ever seen 70-80% OEF in healthy individuals in any scenario. I don’t think you need to change the overall purpose of this paragraph, just rethink the values included.
2. In addition to the changes in PetCO₂ shown in figure 5, please also show absolute P_{Et}CO₂ at each stage, as you have done for other variables.
3. I suggest also showing the changes in blood flow and CVC for RICA and LICA, as you have done with other variables. I notice the ICA change values and comparisons thereof are missing from the results section, while you have included the equivalent comparisons for the BA.
4. I found the results section difficult to read. Since you have the p values in the figures and supplementary tables, why not remove most of them from the results section leaving only those that aren’t presented elsewhere? Reading absolute values, absolute changes, or percentage changes would be preferable to reading the p values.
5. I appreciate that there is a movement towards more lenient approaches to statistical interpretation (Curran-Everett, 2020), and I empathize. As such, I agree with the sentiment that there were differences in body mass and BMI between groups; i.e., the sentence, “There was a trend for body mass (P=0.058) and BMI 293 (P=0.051) to be lower in the group with VAH”, is acceptable. This is fine, given that you go on to give effect size values for these variables. However, leniency in p value interpretation should be explained and supported in the statistical analysis section if that is the case. I say this because the above example of leniency is then quite obviously contrasted in the result section when you consider the p values of 0.074 – 0.0985 from the various ICA comparisons as not significant. Again, whether you have a hard cut off of 0.05 or you accept some degree of tolerance, either way is fine, but you can’t both allow 0.058 and not allow 0.074 without further explaining why. This is a p value difference of 0.016, or 1.6%!

Visually it looks to me that there may have been group differences in size in both the left and right ICA. I’m not sure exactly why this would be the case, but I can also see there being a fairly reasonable anatomical explanation.

I would also would be amenable to interpreting there being an interaction effect on total CBF as a percentage of cardiac output with a p value of 0.071!

In order to more thoroughly motivate the allowance for lenient interpretation of P values please add either evidenced support for this decision, or discussion of the possibility that there may have been false negative findings with the above mentioned variables.

6. In regards to the two previous comments (ICA change comparisons, and stats), it almost looks like the differences in the ICA changes during LBNP between groups may actually add support to your findings (i.e. *without VAH* have less change in ICA flow during LBNP) at least in the RICA.
7. Fig 4C – cardiac index should be l/min/m²

Response to reviewers' comments

We would like to thank the Editors and Reviewers for their careful review of the manuscript. The comments have greatly improved the manuscript. We have addressed each point in turn below.

REFEREE COMMENTS

Referee #1:

Congratulations on a very well conducted study and well-presented manuscript. I have only a small number of comments. See below, in no particular order.

Author response: Thank you for your valuable feedback.

1. Line 486 – “The normal global cerebral oxygen extraction fraction is 15-20% (and may vary by region), with a potential maximum of 70-80%”. I don't believe the paper you reference here measured or reported oxygen extraction fraction, so perhaps find a more appropriate paper to cite. Additionally, I'm not sure that these values are entirely correct. To my knowledge, cerebral OEF is usually around 30%, give or take ~5-10%, and I don't think I have ever seen 70-80% OEF in healthy individuals in any scenario. I don't think you need to change the overall purpose of this paragraph, just rethink the values included.

Author response: Thank you for pointing this out. We apologise for this mistake and have edited the values to 30-40% and have changed the citations. We have also removed maximal potential of 70-80% as we don't think that any study can confirm what the maximal potential OEF is in a healthy human brain.

2. In addition to the changes in PetCO₂ shown in figure 5, please also show absolute PEtCO₂ at each stage, as you have done for other variables.

Author response: Thank you, this has been added.

3. I suggest also showing the changes in blood flow and CVC for RICA and LICA, as you have done with other variables. I notice the ICA change values and comparisons thereof are missing from the results section, while you have included the equivalent comparisons for the BA.

Author response: This has been completed. To be concise we have put a summary sentence and added the change in ICA flow and conductance as a figure to the supplementary material. Figure 2 would have 8 panels on if we

added the change in these variables; which made the figures difficult to look at.

4. I found the results section difficult to read. Since you have the p-values in the figures and supplementary tables, why not remove most of them from the results section leaving only those that aren't presented elsewhere? Reading absolute values, absolute changes, or percentage changes would be preferable to reading the p values.

Author response: We agree and have tried to improve readability. We have removed p-values in the text as suggested.

5. I appreciate that there is a movement towards more lenient approaches to statistical interpretation (Curran-Everett, 2020), and I empathize. As such, I agree with the sentiment that there were differences in body mass and BMI between groups; i.e., the sentence, "There was a trend for body mass ($P=0.058$) and BMI 293 ($P=0.051$) to be lower in the group with VAH", is acceptable. This is fine, given that you go on to give effect size values for these variables. However, leniency in p value interpretation should be explained and supported in the statistical analysis section if that is the case. I say this because the above example of leniency is then quite obviously contrasted in the result section when you consider the p values of 0.074 – 0.0985 from the various ICA comparisons as not significant. Again, whether you have a hard cut off of 0.05 or you accept some degree of tolerance, either way is fine, but you can't both allow 0.058 and not allow 0.074 without further explaining why. This is a p value difference of 0.016, or 1.6%!

Author response: we agree that leniency with p-values needs to be consistent. We have now stated in the statistical analysis section that where p-values were between 0.05 and 0.06 effect sizes have been provided to give an idea of whether there could be an effect size which is physiologically meaningful.

6. Visually it looks to me that there may have been group differences in size in both the left and right ICA. I'm not sure exactly why this would be the case, but I can also see there being a fairly reasonable anatomical explanation.

Author response: Thank you. If we look at the pairwise comparisons for VAH versus the without VAH group, there are no differences in the flow in the left or right ICA at rest ($P=0.1300$ and $P=0.6398$ respectively). There are also no differences between groups at the other LBNP levels between VAH and those without VAH (left ICA at -20 $P=0.3658$, at -40 $P=0.4276$, and tight ICA at -20 $P=0.3535$ and -40 $P=0.0814$).

I would also would be amenable to interpreting there being an interaction effect on total CBF as a percentage of cardiac output with a p value of 0.071! In order to more thoroughly motivate the allowance for lenient interpretation of

P values please add either evidenced support for this decision, or discussion of the possibility that there may have been false negative findings with the above mentioned variables.

Author response: Thank you for pointing this out. We have decided no to interpret this as a trend (given the journals guidelines) and have inserted the fact about possible false negatives in the limitations section.

7. In regard to the two previous comments (ICA change comparisons, and stats), it almost looks like the differences in the ICA changes during LBNP between groups may actually add support to your findings (i.e. without VAH have less change in ICA flow during LBNP) at least in the RICA.

Author response: Thank you for suggesting this. We have alluded to this in the limitations section where we have written about possible false negatives.

8. Fig 4C – cardiac index should be l/min/m²

Author response: Thank you, this has been changed.

Referee #2:

I appreciate the opportunity to review the manuscript entitled "Cerebral blood flow during simulated central hypovolaemia in people with hypertension: does vertebral artery hypoplasia matter?" by Neumann and colleagues.

The present study aimed to explore whether vertebral artery hypoplasia (VAH) in hypertensive patients impacts cerebral blood flow (CBF) regulation during haemodynamic stress (reduced cardiac output and blood pressure) induced via body negative pressure (LBNP). The authors conclude that when compared to hypertensive patients without VAH, patients with VAH evoke a greater systemic peripheral resistance to ultimately preserve CBF during LBNP-induced hypovolemia. This research holds clinical relevance given the prevalence of ischaemic stroke in patients with VAH and the authors deserve commendation for their work in this patient group. Please see below for my constructive comments to the manuscript.

Author response: Thank you for your constructive comments. They have greatly improved the manuscript.

L134: Abbreviate basilar artery to BA here and then amend throughout the manuscript.

Author response: This has been completed.

L150: How was your sample size determined and are the authors confident in their ability to detect subtle differences in their variables of interest?

Author response: This is an important point. We did a sample size calculation before the study based on pilot data where we expected that people with VAH would struggle to maintain their CBF in the face of lower central blood volume. We based this on a partial eta² of 0.11 which gives an effect size f of 0.35. Based on this calculation n=20 participants would provide 80% power to find an interaction effect between LBNP and having VAH (or not). We in fact found the inverse of this where people without VAH had a bigger decrease in CBF. The partial eta² for the interaction effect was 0.15. Since we have a few p-values that are <0.1 but >0.05 for the ICA flows, this suggests that we may not have had enough power to find smaller effect sizes. We have inserted a sentence about false negative rates in the limitations section.

L152: Please include the appropriate reference number for local R&D approval.

Author response: This has been completed.

L156: Please state whether this study was conducted in accordance with the Declaration of Helsinki.

Author response: This has been completed.

L165: You have not defined ABPM.

Author response: This has been completed.

L203: Can the authors confirm the timeframe/order of the cardiac MRI in relation to ICA/BA imaging?

Author response: The cardiac imaging was done at the end of each LBNP level. The flow imaging in the cerebral vessels took ~4 minutes and the aortic flow took 3 mins.

L214: Please include manufacturer details for the LBNP barometer.

Author response: The LBNP was made at the University Hospital Bristol and Weston. We have cited the study showing the design on line 212, page 7 (Ref 16; Hamilton et al 2021. Clin Radiol. 76. 471.e9-471.e16).

L225: Please include manufacturer details for the sphygmomanometer.

Author response: This has been completed.

L237: Please state total CBF here.

Author response: This has been completed.

L240: Please abbreviate total peripheral resistance to TPR.

Author response: This has been completed.

L241: Please include the calculation for cardiac index here and remove from L286.

Author response: This has been completed.

L241: L/min not l/min. Please correct throughout the manuscript.

Author response: This has been completed.

L241: Please abbreviate cardiac output to CO.

Author response: Thank you for the suggestion. We have left as cardiac output as there are many abbreviations already and it limits readability. People with dyslexia struggle to read sentences with too many abbreviations.

L243: mL not ml. Please correct throughout the manuscript.

Author response: This has been completed.

L257: Please provide more detail on "all analyses were blinded".

Author response: Thank you. The person analysing the data was blinded to whether the measurements were done at 0, 20 or 40 mmHg. This has been inserted on line 258.

L273: The results section is rather heavy. Where most variables have been abbreviated, please could you use the abbreviation hence forth to reduce text length.

Author response: We agree that the results are heavy. However, we have not used too many abbreviations as it limits accessibility. With the suggestion of reviewer one to remove in-text p-values, we think the results is now easier to read and more concise. All p-values are in figures or tables.

L273: Within the results section, p-values do not need to be greater than 3 decimal places (as seen in table 1) - please amend where appropriate within the text. The figures are fine having 4 decimal places.

Author response: Thank you for the suggestion. We have now removed in text p-values based on reviewer one comments.

L301: Remove "Bonferroni multiple comparisons". This is stated in the statistical analysis section.

Author response: This has been done.

L317: Units for cerebral vascular conductance should be mL/min/mmHg not mL/mmHg/min.

Author response: We apologise for this mistake and have changed.

L319: Up until this point the authors have provided absolute values or delta values for variables but have not for the remaining results sections. Either add these to the remaining sections or remove them all together.

Author response: We agree with the reviewer. Reviewer 1 also requested that we added this. We originally left this data out of the manuscript because (as this reviewer points out) the results are long. However, we have now added a summary of the changes in haemodynamic variables and ICA flow in the results. The change in haemodynamic variables are in Supplementary Table 2 and the change in flow and conductance for the ICAs are in main figure 2.

L509: The methodological considerations could be written with fewer short sentences. Please remove the values provided in the brackets as it's not clear whether this is a total group observation or a specific group.

Author response: We agree and have improved the wording.

L530: It is not clear what a "Cushing like response" is referring to.

Author response: thank you pointing this out. We have expanded on this point.

L555: Could a funding reference number be added for the Wellcome Trust?

Author response: This was supported by the Wellcome Trust PhD programme. There is not a specific reference number. We have expanded the description.

L680: Table 1. Please confirm the p-values provided for the clinic BP and HR and ABPM are correct as they appear to be identical to each other.

Author response: Thank you for catching this. We apologize for the mistake. We have checked the comparisons and the data for the ABPM and clinic BP. The clinic BP comparisons were not inserted. Table 1 now has the correct p-values and data.

Dear Dr Hart,

Re: JP-RP-2025-287786R1 "Cerebral blood flow during simulated central hypovolaemia in people with hypertension: does vertebral artery hypoplasia matter?" by Sandra Neumann, Jonathan Rodrigues, Lydia L Simpson, Chris B Lawton, Daniel Burden, Matthew D Kobetic, Zoe H. Adams, Katrina Hope, Julian F. R. Paton, Hazel C. Blythe, Nathan E Manghat, Jill N Barnes, Angus K. Nightingale, Mark Hamilton, and Emma C Hart

We are pleased to tell you that your paper has been accepted for publication in The Journal of Physiology.

Yours sincerely,

Harold Schultz
Senior Editor
The Journal of Physiology

If you would like to receive our 'Research Roundup', a monthly newsletter highlighting the cutting-edge research published in The Physiological Society's family of journals (The Journal of Physiology, Experimental Physiology, Physiological Reports, The Journal of Nutritional Physiology and The Journal of Precision Medicine: Health and Disease), please click this link, fill in your name and email address and select 'Research Roundup':
<https://www.physoc.org/journals-and-media/membernews>

- You can help your research get the attention it deserves! Check out Wiley's free Promotion Guide for best-practice recommendations for promoting your work at: www.wileyauthors.com/eeo/guide. You can learn more about Wiley Editing Services which offers professional video, design, and writing services to create shareable video abstracts, infographics, conference posters, lay summaries, and research news stories for your research at: www.wileyauthors.com/eeo/promotion.

EDITOR COMMENTS

Reviewing Editor:

Thank you for addressing the reviewers' comments and for a excellent contribution.

Senior Editor:

The editors wish to thank the authors for these final adjustments to the manuscript. The article is now accepted for publication. Congratulations for an interesting and insightful study. Please consider the Journal of Physiology for your future studies.

REFEREE COMMENTS

Referee #1:

Congratulations on an interesting study and paper.

Referee #2:

I commend the authors on their revised manuscript. They have addressed all of my prior comments to my satisfaction.